# Policy Impacts of High-Standard Farmland Construction on Agricultural Sustainability: Total Factor Productivity-Based Analysis

**Feng Ye [1], Lang Wang [2,*], Amar Razzaq [3]**, **Ting Tong [1], Qing Zhang [1,4]** and **Azhar Abbas [5]**

1   College of Economics and Management, Huazhong Agricultural University, No. 1 Shizishan Street, Wuhan 430070, China
2   College of Business Administration, Zhongnan University of Economics and Law, No. 182 Nanhu Avenue, Wuhan 430073, China
3   Business School, Huanggang Normal University, No. 146 Xingang Second Road, Huanggang 438000, China
4   School of Digital Media and Humanities, Hunan University of Technology and Business, No. 569 Yuelu Avenuet, Changsha 410205, China
5   Institute of Agricultural and Resource Economics, University of Agriculture, Faisalabad, Punjab 38000, Pakistan
\*   Correspondence: wanglang@stu.zuel.edu.cn

**Abstract:** High-standard farmland construction is an important initiative in China that promotes sustainable agricultural development and ensures food security through land consolidation. This study measures the growth of agricultural total factor productivity (ATFP) in China, which is used to characterize the sustainable development of agriculture. Using provincial panel data from China and a continuous difference-in-difference (DID) model, the study examines the impact of high-standard farmland construction policy on ATFP growth. Results show that ATFP in China has an increasing trend with an average annual growth rate of 3.6%. The average enhancement effect of high-standard farmland construction policy on ATFP is 1.0%, which remains significant after various robustness tests. The positive effect of the policy on ATFP becomes apparent in the third year of implementation and shows a gradually increasing trend. The study also finds that the impact of high-standard farmland construction on ATFP is more pronounced in the central regions of China, the main grain-producing regions, and the regions with higher ATFP. High-standard farmland construction policy enhances ATFP by promoting agricultural technology change and technical efficiency. To promote the growth of ATFP and achieve sustainable agricultural development, China should continue to promote the construction of high-standard farmland and explore suitable construction models for different regions.

**Keywords:** rural–urban migration; rural development; land consolidation; income distribution

## 1. Introduction

Promoting sustainable agriculture and meeting the global demand for food is a major challenge for humanity [1,2]. Increasing agricultural total factor productivity (ATFP) is crucial for promoting sustainable agricultural development [3]. Previous research has shown that a key factor in the sustained growth of Chinese agriculture is the increase in ATFP [4–6]. ATFP is the portion of agricultural output that is not explained by the inputs used for production [7]. As ATFP represents the ability of creation to grow under given input conditions, the level of ATFP is an important basis for assessing the sustainability of agriculture [8–10].

The Chinese government places great importance on improving ATFP. In recent years, China has been promoting a policy of building high-standard farmland. High-standard farmland construction is an agricultural land improvement project that aims to make farmland concentrated, flat, high-yielding, and ecologically improved by improving farmland infrastructure [11,12]. Theoretically, high-standard farmland construction can improve

farmland quality, promote agricultural scale operation, and thus increase ATFP [13,14]. However, in reality, the impact of high-standard farmland construction policy on ATFP is not yet known due to the geographical differences among provinces and the quality of policy implementation. The objective of this paper is to assess the impact of high-standard farmland construction on ATFP using an econometric approach, with the aim of providing a new perspective for sustainable agricultural development.

The method for measuring ATFP is complex. There are two main methods for calculating ATFP: stochastic frontier analysis (SFA) and data envelopment analysis (DEA). SFA is a parametric estimation method that involves setting specific functional forms and probability distributions for random error terms [15,16]. DEA, on the other hand, is a nonparametric estimation method that calculates efficiency by enveloping the production frontier [17]. Many studies use a combination of DEA and the Malmquist index to measure ATFP [18,19]. As ATFP measurement methods have improved, scholars have started to focus on the determinants of ATFP. Improving farmers' human capital enables them to adopt more advanced technology, which significantly improves ATFP [20]. Infrastructure development can improve agricultural production conditions and increase the land strength of cultivated land, contributing to the advancement of ATFP [21,22]. Agricultural subsidies can help farmers alleviate financial constraints in agricultural production and invest more or adopt more advanced production technologies, thereby increasing ATFP [23,24]. Agricultural and institutional reforms, such as the household responsibility system, agricultural taxation reform, and land system reform in China, have also played a significant role in the growth of ATFP in China [25–28].

Previous research focused on the factors that influence ATFP growth, such as human capital, infrastructure development, government investment, and agricultural policy changes [29,30]. Among these, agricultural policy changes and infrastructure development are particularly important factors in influencing ATFP. In China, a high-standard farmland construction policy is a government policy aimed at improving agricultural infrastructure. The Chinese government invested heavily in the construction of high-standard farmland. Previous research focused on the impact of this policy on farmers' income and eco-efficiency [31,32] but neglected its impact on agricultural sustainability. This paper aims to address this gap by using ATFP to assess the impact, heterogeneity, and mechanisms of high-standard farmland construction policy on agricultural sustainability in China.

This study makes four contributions to the literature. First, we examine the causal relationship between land reclamation and ATFP based on the construction of high-standard farmland in China, offering a new perspective on sustainable agricultural development. Second, this paper investigates the heterogeneous effects of policy implementation on ATFP from multiple angles, providing empirical evidence for the promotion and improvement of high-standard farmland construction policies. Third, this study's continuous difference-in-difference-based research design effectively addresses the endogeneity problem of policy change, accurately identifying the causal relationship between high-standard farmland policy and ATFP. Fourth, this research provides empirical evidence for developing countries to promote the construction of high-standard farmland and sustainable agricultural development.

## 2. Policy Background

The construction of high-standard farmland is an important component of land consolidation in China, which aims to promote sustainable agricultural development and ensure food security. According to the document "Standard for Construction of High-Standard Basic Farmland", high-standard farmland is defined as "Basic farmland formed through rural land remediation that is concentrated and contiguous, with supporting facilities, high and stable yields, good ecology, strong disaster resistance, and compatible with modern agricultural production and operation methods." The construction of high-standard farmland in China can be divided into two phases: the exploration phase (1988–2010) and the standardized implementation phase (2011–present).

Before 2011, there were no professional documents specifying the measurement standards and construction requirements for high-standard farmland. During this phase, the main focus of comprehensive land development was on increasing the area of arable land. In 2011, the Chinese government launched the National Land Improvement Plan (2011–2015), which established the construction standards and requirements for high-standard farmland. Local governments also formulated their own guidelines for implementing high-standard farmland based on the national document, marking the start of a standardized period for high-standard farmland in China. In this paper, data from 2013 and 2017 were selected for visual analysis because they are the years of advancement of high standards of construction, and the results are shown in Figure 1.

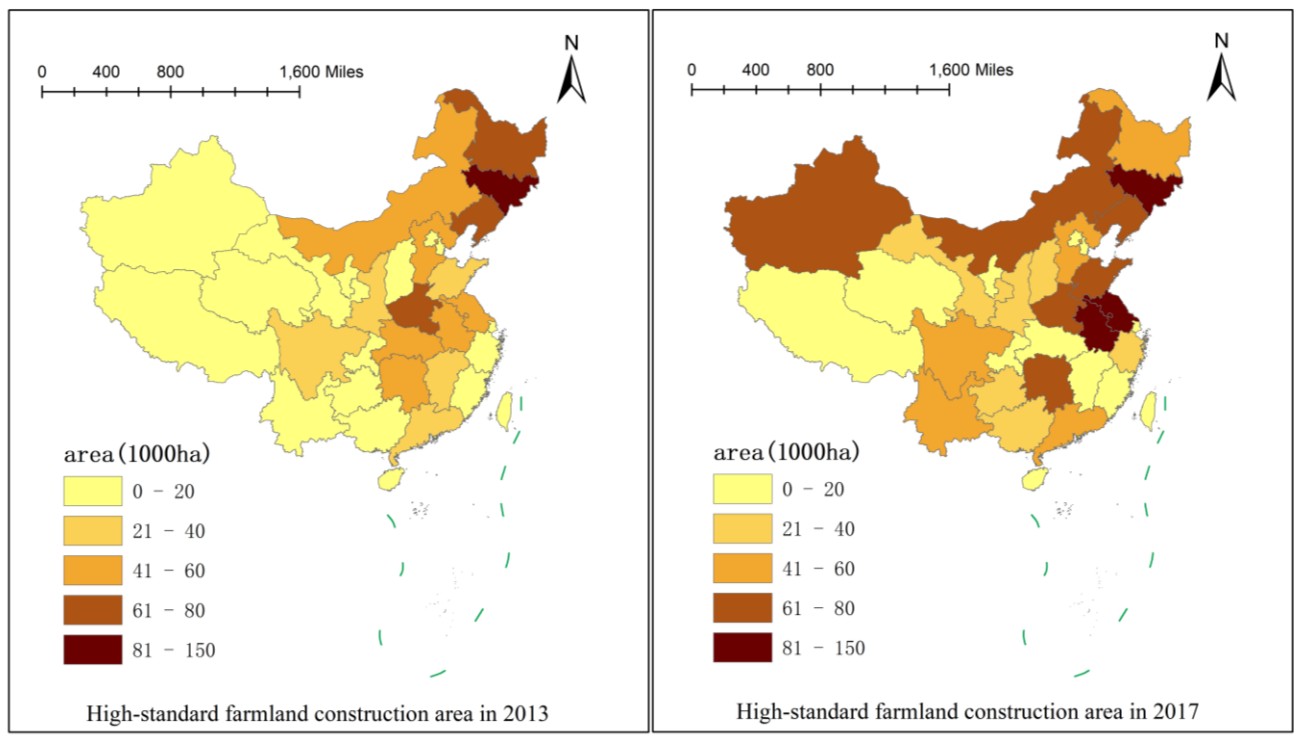

**Figure 1.** Change in the new area of high-standard farmland.

After 2013, high-standard farmland entered the stage of large-scale standardized construction, and the scope of construction gradually expanded to 31 provinces in China. In 2013, 99,000 hectares of high-standard farmland were constructed, with the top three provinces in terms of construction area being Jilin, Heilongjiang, and Henan, with 92,070 hectares, 71,470 hectares, and 69,470 hectares of high-standard farmland constructed, respectively, according to *China Financial Statistics Yearbook*. By 2017, China had accumulated a total of 46.67 million hectares of high-standard farmland construction, with Shandong, Henan, and Jiangsu accumulating over 3 million hectares of high-standard farmland construction each, according to *China Financial Statistics Yearbook*.

## 3. Model and Data

### 3.1. Model

#### 3.1.1. Total Factor Productivity Measurement Model

In order to accurately measure ATFP in this paper, we adopt the DEA method and combine it with the Super Efficiency model and the global Malmquist index proposed by Pastor and Lovell [33] and Oh [34]. This hybrid approach, known as the EBM–SGM index and proposed by Tone and Tsutsui [35], is used to construct the production frontier. The EBM–SGM index considers both radial and non-radial slack variables and avoids

the defects of linear programming non-solution and non-transmissibility. To calculate the EBM–SGM index, we use the following formula:

$$r^* = \min\theta - \varphi \sum_{i=1}^{m} \frac{\omega_i s_i}{m_0}$$
$$\text{s.t.} \{\theta m_0 - M\rho - s = 0; \rho N \geq n_0; \rho \geq 0, s \geq 0\} \tag{1}$$

In Equation (1), $r^*$ represents the production efficiency value, $\theta$ represents the radial efficiency value, and $\varphi$ represents the parameter considering both radial and non-radial slack variables. $w_i$ is the relative importance of the i factor of production and $s_i$ is the slack variable of the i factor of production. $\rho$ is the relative weight, and M and N represent the input and output vectors, respectively. $m_0$ and $n_0$ represent the input and output levels under the radial constraint, respectively.

In order to accurately measure ATFP in this study, the EBM Super-Global–Malmquist (EBM–SGM) index is used. This combination effectively avoids issues such as linear programming non-solution and non-transmissibility. The formula for the EBM–SGM index is shown below.

$$\text{ATFP}^{t,t+1}\left(m^t, n^t; m^{t+1}, n^{t+1}\right) = \left[\frac{1+D^t\left(m^t,n^t\right)}{1+D^t\left(m^{t+1},n^{t+1}\right)} \times \frac{1+D^{t+1}\left(m^t,n^t\right)}{1+D^{t+1}\left(m^{t+1},n^{t+1}\right)}\right]^{\frac{1}{2}}$$
$$= \frac{1+D^t\left(m^t,n^t\right)}{1+D^t\left(m^{t+1},n^{t+1}\right)} \times \left[\frac{1+D^{t+1}\left(m^t,n^t\right)}{1+D^t\left(m^t,n^t\right)} \times \frac{1+D^{t+1}\left(m^{t+1},n^{t+1}\right)}{1+D^t\left(m^{t+1},n^{t+1}\right)}\right]^{\frac{1}{2}} \tag{2}$$
$$= \text{TE}\left(m^{t+1}, n^{t+1}; m^t, n^t\right) \times \text{TC}\left(m^{t+1}, n^{t+1}; m^t, n^t\right)$$

In Equation (2), $D_t$ and $D_{t+1}$ denote the set of production technologies in periods t and t + 1, respectively. Referring to Färe et al. [36], ATFP can be decomposed into agricultural technical change (TC) and agricultural technical efficiency (TE).

### 3.1.2. Model of the Impact of High-Standard Farmland Construction Policy on ATFP

Referring to the existing literature [37], this paper uses a continuous DID model to estimate the effect of the high-standard farmland policy on ATFP. The following model is constructed in this paper based on this analysis.

$$\text{LNATFP}_{i,t} = \alpha_1 + \alpha_2 \text{treated}_i \times \text{time}_t + \beta X_{i,t} + \eta_t + \gamma_i + \mu_{i,t} \tag{3}$$

In Equation (3), i stands for region and t stands for year. ATFP stands for agricultural total factor productivity, treated stands for proportion of high-standard farmland. $\text{time}_t$ stands for the dummy variable at the time of policy. When $t \geq 2011$, time is 1; otherwise, it is 0. X stands for control variable, $\gamma_i$ and $\eta_t$ stand for year effect and region effect, respectively. $\mu_{i,t}$ stands for classical random disturbance term. $\alpha$ and $\beta$ stand for parameters to be estimated. In particular, it is important to note that $\alpha_2$ is the core estimated parameter in this paper, representing the net effect of high-standard farmland construction policy on ATFP.

### 3.1.3. Parallel Trend Test

The parallel trend assumption is a crucial prerequisite for DID estimation. Based on previous research [38], the following model is constructed in this paper to test the parallel trend assumption.

$$\text{LNATFP}_{i,t} = \sum_{k=2008}^{2017} \beta_k \text{treated}_i \times d_t + \beta X_{i,t} + \eta_t + \gamma_i + \mu_{i,t} \tag{4}$$

Equation (4) shows this model, where time represents the year dummy variable, treated represents the area of high-standard farmland construction, and other variables and coefficients are set consistently with Equation (3). We can determine whether the parallel trend assumption holds by examining the statistical significance of the estimated parameters of the interaction term. If the estimated parameters of the interaction term are statistically insignificant before 2011, we can assume that the parallel trend assumption is valid.

3.1.4. Impact Mechanism Model

This paper will study the impact of high-standard farmland construction policies on total factor productivity (ATFP) by decomposing ATFP into technical change (TC) and technical efficiency (TE). This approach was used in previous studies [39,40]. The impacts on TC and TE will be analyzed separately to understand the mechanism behind the effects of these policies. Two models will be used for this analysis:

$$\text{LNTC}_{i,t} = \alpha_1 + \alpha_2 \text{treated}_i \times \text{time}_t + \beta X_{i,t} + \eta_t + \gamma_i + \mu_{i,t} \tag{5}$$

$$\text{LNTE}_{i,t} = \alpha_1 + \alpha_2 \text{treated}_i \times \text{time}_t + \beta X_{i,t} + \eta_t + \gamma_i + \mu_{i,t} \tag{6}$$

The estimated parameters of the $\text{treated}_i \times \text{time}_t$ variable in Equations (5) and (6) represent the effects of high-standard farmland construction policy on TC and TE, respectively.

*3.2. Variables*

3.2.1. Explained Variables

In this study, ATFP is the main explained variable. To calculate ATFP, we use the EBM–SGM method, which requires the selection of appropriate input and output variables. According to existing research [41–43], the input variables chosen in this study include: (1) the combined agricultural sown area and aquaculture area, (2) the number of people employed in the primary industry at the end of the year, (3) the total power of agricultural machinery, (4) the amount of fertilizer, and (5) the effective irrigation area in agriculture. These variables represent land, labor, machinery, fertilizer, and irrigation inputs in agricultural production. The total agricultural output value is used as the agricultural output indicator in this study.

3.2.2. Core explanatory Variable

The key explanatory variable in this paper is the policy variable $\text{treated}_i \times \text{time}_t$ (difference-in-differences) of high-standard farmland construction. This variable is created through the interaction of a time dummy variable, representing the implementation of the policy and the regional area of high-standard farmland treated. The size of its parameter indicates the impact of the high-standard farmland construction policy on ATFP.

3.2.3. Control Variables

According to existing research [44,45], the following variables are used as control variables in this study: (1) Infrastructure, represented by the number of road miles per unit area (ROAD); (2) Human capital, represented by the average number of years of education for the regional labor force (EDU); (3) Urbanization level, represented by the ratio of the urban population to the total population (UR); (4) Land quality, represented by the ratio of effective irrigated area to sown area (LAQA); (5) Disaster rate, represented by the ratio of the disaster area to total sown area (DR) to control for the impact of climate on ATFP; (6) Agricultural planting structure, represented by the ratio of sown area of food crops to the total sown area (AS); (7) Fiscal support to agriculture, represented by the ratio of fiscal support for agriculture expenditure to total fiscal expenditure (AF). These variables are included to account for their potential influence on ATFP and to ensure the accuracy of the results of this study.

*3.3. Data*

This paper uses data from 30 provinces in China for the study period of 2008–2017. Hong Kong, Macau, Taiwan, and Tibet are not included due to a lack of sufficient statistical data. The data used for calculating ATFP, including input and output variables, were obtained from the *China Statistical Yearbook* and *China Rural Statistical Yearbook*. Control variables were obtained from the *China Statistical Yearbook* and EPS database. Data on high-standard farmland construction was obtained from the *China Financial Yearbook*. Some

abnormal data were removed, and missing data were completed using the interpolation method. The statistical description of the data used in this paper is shown in Table 1.

**Table 1.** Descriptive statistics of variables.

| Variables | Abbreviation | Units | $N$ | Mean | S.D. | Min | Max |
|---|---|---|---|---|---|---|---|
| Agricultural total factor productivity | ATFP | - | 300 | 1.037 | 0.055 | 0.886 | 1.210 |
| Agricultural technology change | TC | - | 300 | 1.035 | 0.041 | 1.000 | 1.205 |
| Agricultural technology efficiency | TE | - | 300 | 1.003 | 0.052 | 0.862 | 1.210 |
| $\text{treated}_i \times \text{time}_t$ | DID | - | 300 | 2.544 | 1.909 | 0.000 | 5.782 |
| Total agricultural production value | Output | Billion Yuan | 300 | 232.8 | 168.7 | 14.5 | 804.3 |
| Land input | Input1 | 1000 km$^2$ | 300 | 5687.0 | 3808.0 | 123.9 | 15,205.0 |
| Labor input | Input2 | 10,000 individuals | 300 | 939.5 | 666.6 | 37.1 | 2847.0 |
| Mechanical input | Input3 | 10,000 kW | 300 | 3257.0 | 2923.0 | 95.3 | 13,353.0 |
| Fertilizer input | Input4 | 10,000 tons | 300 | 191.6 | 146.1 | 8.0 | 716.1 |
| Irrigation inputs | Input5 | 1000 km$^2$ | 300 | 2096.0 | 1570.0 | 115.5 | 6031.0 |
| Infrastructure | ROAD | Km | 300 | 0.915 | 0.506 | 0.079 | 2.297 |
| Human capital | EDU | Year | 300 | 9.616 | 1.143 | 6.971 | 13.530 |
| Urbanization level | UR | % | 300 | 0.547 | 0.132 | 0.291 | 0.896 |
| Land quality | LAQA | % | 300 | 0.389 | 0.183 | 0.118 | 1.000 |
| Disaster rate | DR | % | 300 | 0.195 | 0.138 | 0.000 | 0.695 |
| Agricultural planting structure | AS | % | 300 | 0.654 | 0.130 | 0.353 | 0.958 |
| Fiscal support to agriculture | AF | % | 300 | 0.111 | 0.030 | 0.030 | 0.190 |

## 4. Empirical Results

### 4.1. The Results of ATFP Measurement

In this study, the Empirical Border Malmquist Super-Global index was used to measure the Total Factor Productivity (ATFP) growth in China from 2008 to 2017 (Table 2). The results show that China's ATFP experienced an upward trend with an average annual growth rate of approximately 3.6% during this period. Most provinces had positive ATFP growth, and the growth of ATFP was generally balanced across provinces. The results also show that the growth of ATFP in China was determined by a combination of technological change (TC) and technical efficiency (TE). The average annual growth rate of TC was 2.8%, while the average annual growth rate of TE was 0.7%. These findings are consistent with those of previous studies and by existing studies [39,46,47]. The Malmquist index was converted to a growth index with 2008 as the base period to understand the results better.

**Table 2.** ATFP Growth and Decomposition in China from 2008 to 2017.

| Year | TFP | TC | TE |
|---|---|---|---|
| 2008–2009 | 0.946 | 1.022 | 0.927 |
| 2009–2010 | 1.128 | 1.087 | 1.039 |
| 2010–2011 | 1.030 | 1.020 | 1.010 |
| 2011–2012 | 1.051 | 1.074 | 0.980 |
| 2012–2013 | 1.039 | 1.072 | 0.966 |
| 2013–2014 | 1.040 | 1.042 | 0.999 |
| 2014–2015 | 1.032 | 1.009 | 1.023 |
| 2015–2016 | 1.053 | 1.010 | 1.043 |
| 2016–2017 | 1.054 | 1.011 | 1.042 |
| Mean | 1.036 | 1.028 | 1.007 |

Note: The mean values in Table 2 are calculated from the geometric mean.

### 4.2. Baseline Regression Results

The effects of the high-standard farmland construction policy on ATFP were analyzed in this section, with the results shown in Table 3. Model 1 represents the results without controlling for any variables, while Model 2 shows the results after adding control variables. The results in Table 3 indicate that, when controlling for all variables, time-fixed effects and regional-fixed effects, the impact of the high-standard farmland construction policy

on ATFP is significant at the 1% confidence level with an estimated coefficient of 0.010. This indicates that, on average, the policy has increased ATFP by 1.00%. The policy reform had a significant effect on ATFP. As a key land remediation project in China, the construction of high-standard farmland can significantly improve ATFP and promote sustainable agricultural development.

**Table 3.** Baseline regression results.

| Variables | Model 1 | Model 2 |
|---|---|---|
| $\text{treated}_i \times \text{time}_t$ | 0.545 *** (0.003) | 0.010 *** (0.004) |
| ROAD | — | 0.274 *** (0.080) |
| EDU | — | 0.516 *** (0.140) |
| UR | — | 0.536 *** (0.110) |
| LAQA | — | 0.261 *** (0.051) |
| AS | — | −0.043 (0.146) |
| AF | — | 0.207 (0.356) |
| DR | — | −0.012 (0.039) |
| Time fixed effects | Yes | Yes |
| Regional fixed effects | Yes | Yes |
| _Cons | 4.624 *** (0.009) | 1.090 *** (0.412) |
| $R^2$ | 0.514 | 0.743 |
| N | 300 | 300 |

Note: *** is significant at the significance level of 1%.

According to the results, infrastructure development can significantly increase ATFP, which is consistent with the findings of Shamdasani et al. [22]. Human capital, represented by the average number of years of education for the regional labor force, can also promote ATFP growth, which is consistent with the study of Chen et al. [47]. Farmers with higher human capital are more capable of adopting new agricultural technologies and are more likely to be familiar with them. The level of urbanization can significantly increase ATFP, which is generally consistent with the results of Li et al. [42]. Land quality has a significant positive effect on ATFP growth, which is generally consistent with the findings of Song and Pijanowski [48]. In addition, the regression results of disaster rate, agricultural restructuring coefficient, and fiscal support to agriculture are statistically insignificant, and this paper has yet to find empirical evidence that disaster rate, agricultural restructuring, and fiscal support to agriculture will enhance ATFP.

### 4.3. Dynamic Effect of the Policy

To further understand the policy's impact, this paper explores the policy's dynamic effects on ATFP through an interaction regression of the policy variable and time dummy variables [37]. The regression results are shown in Table 4. Model 1 controls for time and area effects, while Model 2 adds control variables based on Model 1. The results in Table 4 suggest the following conclusions. First, the estimated parameters for the first two years of policy implementation are statistically insignificant, indicating that the policy's impact on ATFP growth has a lag period of two years. Second, the estimated parameters increase with increasing years, indicating that the policy's effect on ATFP growth is continuous and increasing. We believe that there are differences in the understanding of the policy of high-standard farmland construction in various regions and no standardized implementation, which leads to the lagging effect of the policy.

**Table 4.** Dynamic effects of the policy.

| Variables | Model 1 | Model 2 |
|---|---|---|
| Policy × 2011 | 0.024 *** | 0.001 |
| | (0.005) | (0.005) |
| Policy × 2012 | 0.035 *** | 0.004 |
| | (0.005) | (0.005) |
| Policy × 2013 | 0.042 *** | 0.016 *** |
| | (0.004) | (0.005) |
| Policy × 2014 | 0.048 *** | 0.021 *** |
| | (0.004) | (0.005) |
| Policy × 2015 | 0.053 *** | 0.023 *** |
| | (0.004) | (0.005) |
| Policy × 2016 | 0.063 *** | 0.027 *** |
| | (0.004) | (0.005) |
| Policy × 2017 | 0.077 *** | 0.037 *** |
| | (0.004) | (0.006) |
| Control variables | No | Yes |
| Time fixed effects | Yes | Yes |
| Regional fixed effects | Yes | Yes |
| _Cons | 4.634 *** | 2.560 *** |
| | (0.008) | (0.431) |
| $R^2$ | 0.659 | 0.784 |
| N | 300 | 300 |

Note: *** indicates significance at the significance level of 1%.

### 4.4. Heterogeneity Analysis

The impact of high-standard farmland construction policy on ATFP is influenced by resource endowment and policy bias. There is regional heterogeneity in the effects of ATFP. The results of heterogeneity can better optimize the policy. In this paper, heterogeneity is analyzed based on natural geographical location, agricultural functional areas, and productivity differences.

#### 4.4.1. Heterogeneity of Natural Geographic Location

We consider that there will be significant heterogeneity in the impact of high-standard farmland construction policies on ATFP growth in different natural geographic locations. This study divides the study area into three regions in eastern, central, and western China and conducts grouped regressions based on Equation (3). The results, shown in Table 5, suggest that in the central region of China, the construction of high-standard farmland has the most significant effect on improving ATFP. However, the results for the samples from the eastern and western regions are not statistically significant.

**Table 5.** Results of heterogeneity of natural geographic location.

| Variables | Model 1 | Model 2 | Model 3 |
|---|---|---|---|
| treated$_i$ × time$_t$ | −0.008 | 0.028 *** | −0.005 |
| | (0.010) | (0.007) | (0.006) |
| Control variables | Yes | Yes | Yes |
| Time fixed effects | Yes | Yes | Yes |
| Regional fixed effects | Yes | Yes | Yes |
| _Cons | −0.410 | 1.286 | −0.623 |
| | (0.751) | (0.797) | (0.759) |
| $R^2$ | 0.834 | 0.835 | 0.816 |
| N | 90 | 90 | 120 |

Note: *** indicates significance at the significance level of 1%.

#### 4.4.2. Productivity Heterogeneity

For provinces with different productivities, the effect of high-standard farmland construction on ATFP may vary. In this paper, the policy effects on different quartiles

of ATFP are assessed with the help of panel unconditional quantile regressions, and the results are shown in Table 6. Compared with conditional quantile regression, unconditional quantile regression does not depend on the increase or decrease of control variables and is widely used for heterogeneity analysis of treatment effects [49]. Model 1, Model 2, Model 3, and Model 4 represents the regression results for quartiles 25, 50, 75, and 90, respectively. The estimated coefficients were 0.041 and 0.026 at quintiles 25 and 50, respectively, and were statistically significant. The estimated coefficients were not significant at the 75th and 90th quartiles. The study's results indicated that the effect of high-standard farmland construction on ATFP gradually diminished as the quantile increased. High-standard farmland as a land improvement policy can significantly reduce the differences in ATFP among provinces.

**Table 6.** Unconditional quantile regression results.

| Variables | Model 1 | Model 2 | Model 3 | Model 4 |
|---|---|---|---|---|
| $treated_i \times time_t$ | 0.041 *** | 0.026 ** | −0.007 | −0.020 |
| | (0.011) | (0.012) | (0.014) | (0.013) |
| Control variables | Yes | Yes | Yes | Yes |
| Time fixed effects | Yes | Yes | Yes | Yes |
| Regional fixed effects | Yes | Yes | Yes | Yes |
| _Cons | 1.924 ** | 0.939 | −1.066 | 0.210 |
| | (0.775) | (0.889) | (1.491) | (2.361) |
| $R^2$ | 0.338 | 0.605 | 0.416 | 0.204 |
| N | 300 | 300 | 300 | 300 |

Note: *** and ** indicate significance at the significance level of 1% and 5%.

## 5. Discussion

During 2008–2017, China's ATFP showed an increasing trend, which is similar to the findings of Liu et al. [50] and Li and Lin [41]. The decomposition of the ATFP shows that China's ATFP growth is still largely dependent on technological change. This finding is generally consistent with that of Fan et al. [51]. Unlike these studies, the focus of our study is to explain the possible reasons for the changes in ATFP in China, which can help us to understand the sustainable growth of Chinese agriculture. This paper argues that the improvement of China's ATFP is mainly due to the following aspects: first, China has continued to make efforts to improve the quality of agricultural science and technology in recent years, breaking through several critical technological bottlenecks and applying several high-tech achievements, such as breeding and promoting important new varieties of super rice, water-saving and drought-resistant wheat, and transgenic insect-resistant cotton [52,53]. Secondly, China has invested more in agricultural infrastructure in recent years, improving agricultural production conditions and significantly contributing to ATFP growth [54].

Then, our research shows that a high-standard farmland construction policy can significantly improve ATFP and thus promote sustainable agricultural development. Existing studies started to discuss the impact of high-standard farmland construction on farmers' behavior and farm income [31,32] but lacked a discussion on sustainable agricultural development. We innovatively used ATFP to measure agricultural sustainability and studied the policy effects. Our study can provide lessons for sustainable agricultural development in developing countries. We think the main reasons why high-standard farmland construction can contribute to ATFP growth are as follows. First, the construction of high-standard farmland can enhance agricultural production conditions and increase the disaster resistance of agriculture, thus ensuring food security. In agricultural production, irrigation has always been the weak segment. Traditional agriculture often reduces grain yield due to untimely or insufficient irrigation [55–57]. The construction of high-standard farmland can effectively alleviate the problem of difficult irrigation and promote the improvement of agricultural ATFP. Secondly, high-standard farmland enhances agricultural scale, mechanization, and social services and supports agricultural transformation and upgrading. In China, the problem of land fragmentation is severe, which is not conducive to agricultural

scale and mechanization. A study by Adamopoulos and Restuccia [58] showed that low agricultural productivity in developing countries mainly comes from small planting scale, low productive investment, and low agricultural mechanization. High-standard farmland can effectively compensate for these deficiencies and thus increase ATFP. In addition, our study shows that the policy effects of high-standard farmland are progressively growing, further suggesting that the construction of high-standard farmland can promote sustained agricultural productivity growth.

Moreover, our research indicates that the effects of high-standard farmland construction policy on ATFP are significantly heterogeneous under different conditions, which is also an important contribution to our paper. Through heterogeneity analysis, we can obtain the differences in policy effects in different regions or different groups. The results of heterogeneity can help us optimize the high-standard farmland construction policy in a more targeted way. First, the high-standard farmland construction policy significantly affects the central region rather than the east and west. This paper considers the possible reason for this is that central China is the main grain-producing region, and the construction of high-standard farmland is more standardized [59]. Second, the enhancement effect of the high-standard farmland construction policy is more obvious in the main grain-producing regions. The reason is that the construction of high-standard farmland in China mainly focuses on grain production [60]. Finally, the high-standard farmland construction policy is more pronounced in provinces with lower ATFP, which suggests that high-standard farmland construction is an important policy tool to reduce inter-regional productivity differences and may narrow the income gap between provinces. In addition, high-standard farmland construction policies can increase ATFP through technological change and technical efficiency improvements. The results suggest that high-standard farmland enhancement of agricultural ATFP is multi-dimensional. After the construction of high-standard farmland, farmers can introduce more advanced agricultural technologies into agricultural production, which will significantly promote agricultural technological change [61,62]. At the same time, the construction of high-standard farmland will reduce the degree of land fragmentation, which can improve the efficiency of agricultural technology [32].

Based on our findings, we recommend the following actions: firstly, continue promoting the construction of high-standard farmland. China's current proportion of high-standard farmland is still low and requires further efforts to strengthen construction efforts. To support this, the government should increase financial investment in the construction of high-standard farmland and consider implementing balanced matching funds from both the central and local governments. Additionally, the construction area of high-standard farmland should be carefully planned, and the overall area of high-standard farmland should be increased to support sustainable agricultural development. Then, improve the quality and standards of high-standard farmland construction. The government should prioritize quality management during project implementation, refine construction quality requirements, and ensure that construction units follow technical specifications closely. Additionally, the acceptance process for high-standard farmland projects should be thorough and rigorous. Finally, explore local solutions for high-standard farmland construction. Local governments should consider developing differentiated policies for constructing high-standard farmland based on the heterogeneity of policy implementation in different regions. For example, in the eastern region, agricultural science and technology research, development, and promotion should be prioritized. In the central region, water-saving irrigation technology should be promoted to improve water resource utilization efficiency. In the western hilly areas, the promotion of local practical technologies should be emphasized.

## 6. Conclusions and Prospects

### 6.1. Conclusions

In this study, we employed China's high-standard farmland construction policy as a "quasi-natural experiment" to investigate the relationship between land consolidation and agricultural sustainability. We used Agricultural Total Factor Productivity (ATFP) as a

measure of sustainable agricultural development and employed a continuous difference-in-differences (DID) model to identify the causal relationship between the policy and ATFP and to explore the dynamic effects and mechanisms at play. Our findings indicate that a high-standard farmland construction policy can improve ATFP and thus promote sustainable agricultural development in China. Specifically, we found that:

(1) ATFP in China demonstrated an upward trend during the period 2008–2017, with an average annual growth rate of 3.6%. This growth was driven by technological change and technical efficiency improvement, with an average annual growth rate of 2.8% for technological change and 0.7% for technical efficiency.

(2) The high-standard farmland construction policy had an average effect of 1.0% on ATFP, a result that was robust to a series of robustness tests. The effect of the policy on ATFP was time-heterogeneous, with the effect appearing only in the third year of policy implementation and showing a gradually increasing trend.

(3) The improvement of ATFP by high-standard farmland construction policies has obvious regional heterogeneity. The effect of the policy on ATFP improvement is more pronounced in central China and in provinces with higher ATFP levels.

(4) The policy improved ATFP by promoting technological change and technical efficiency improvement. The policies improve technical change by 1.3% and technical efficiency by 1.4%, and both are statistically significant at the 1% level.

### 6.2. Research Limitations and Prospects

Our study provides new evidence to promote sustainable agricultural development in China. However, the article still has some limitations. First, limited by the availability of data, our study data are only updated to 2017, and future data updates are needed for further research. Second, we only measured agricultural sustainability from the perspective of ATFP, and future research can study the impact study of high-standard farmland construction policy on agricultural sustainability from the perspective of green efficiency.

**Author Contributions:** Conceptualization, F.Y. and L.W.; methodology, F.Y. and L.W.; software, F.Y.; validation, F.Y., L.W. and T.T.; formal analysis, F.Y., A.R. and L.W.; investigation, F.Y.; resources, F.Y. and L.W.; data curation, F.Y. and L.W.; writing original draft preparation, F.Y., L.W. and A.R.; writing—review and editing, F.Y., L.W., A.R., Q.Z, A.A. and T.T.; visualization, F.Y., T.T. and A.A.; supervision, L.W. and Q.Z.; project administration, L.W. and Q.Z.; funding acquisition, L.W. All authors have read and agreed to the published version of the manuscript.

**Funding:** This work was supported by the National Social Science Foundation of China (Project No. 18ZDA072); The Study on the Influence of Toilet Reform Support Policy on Rural Residential Environment Improvement: A Case Study of Changzhutan Area, Hunan Province (Project No. 2022[174]); "You Xiang Jia" APP—Sharing Interactive Platform for Rural Culture and Tourism (Project No. [2021]13).

**Institutional Review Board Statement:** Not applicable.

**Informed Consent Statement:** Not applicable.

**Data Availability Statement:** The datasets used and analyzed during the current study are available from the corresponding author upon reasonable request.

**Conflicts of Interest:** The authors declare no conflict of interest.

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
