# Peer review of "Policy Impacts of High-Standard Farmland Construction on Agricultural Sustainability: Total Factor Productivity-Based Analysis"

_land, doi:10.3390/land12020283_

Round 1

Reviewer 1 Report

land-2151073: Policy Impacts of High-Standard Farmland Construction on Agricultural Sustainability: Total Factor Productivity-based Analysis

Recommendation: MAJOR Revisions

Comments to the Author:

This study measures the growth of agricultural total factor productivity (ATFP) in China, which is used to characterize the sustainable development of agriculture. Using provincial panel data from China and a continuous difference-in-difference (DID) model, the study examines the impact of high-standard farmland construction policy on ATFP growth. This paper examines a significant issue in China, and the authors also use some sound empirical methods to analyze the problem, finding some useful results and corresponding policy implications. The writing is mostly clear, and the results are meaningful. However, some issues need to be solved. I have some comments, which I hope will be helpful for authors.

(1) The introduction is well organized. Starting with the background of the study, the author explains the importance of the research topic and then lists the contributions of the article. However, the research objectives of this paper are not highlighted in the introduction, and the main research objectives of this paper need to be added in the appropriate place.

(2) The authors used Chinese provincial-level data from 2008-2017 to analyze the impact of high-standard farmland construction policy on total factor productivity growth in agriculture. The data use is appropriate. However, some areas deserve further improvement. Some data are listed in article 2.2, but the authors do not indicate the source of the data, and the authors need to indicate the source of the data further. Then, in Figure 1, further explanation is needed as to why the authors selected only two years of data from 2013 and 2017 for the visualization analysis.

(3) The authors used the more cutting-edge of DEA methods to measure total factor productivity in agriculture, which set the stage for the study. However, there are still areas that need to be enhanced. The most important point when calculating total factor productivity is the selection of input-output indicators, and the authors should cite more literature here to support the selection of indicators. What are the advantages of using the (EBM-SGM) index to measure ATFP and why did the authors choose this method instead of other methods to measure it.

(4) The article uses several econometric methods to study the effects of high-standard farmland policies and comes up with some rather enlightening conclusions. However, the authors may need to add some economics discussion to the statistics. In 4.3, the authors' conclusions suggest a lag in the impact of high-standard farmland construction on ATFP, and the authors need to explain the reasons for the existence of the lag.

(5) The article explores policy heterogeneity in several dimensions, giving some rather interesting results that enrich the study. However, one question worth pondering is the authors' purpose in doing all this heterogeneity, and what methods were used to achieve it.

(6) In Table 2, the authors calculate the mean value of ATFP for China for all years, and it is necessary to indicate in the notes below the table whether the geometric or arithmetic mean is used.

(7) The author's discussion section is more than adequate. However, some necessary literature citations are lacking in the last paragraph of the discussion, which leads to a decrease in the article's credibility, and I hope the authors add some highly relevant literature.

(8) The article uses econometric methods to obtain more interesting conclusions. However, the authors need to improve on the writing style of the study conclusions. The conclusions in the abstract and the final study conclusions cannot be identical and need to be differentiated. The final conclusion is an extension and enhancement of the abstract conclusion.

(9) The article has some formatting issues, such as unclear formula (3) and incorrect formatting in the policy recommendation section.

(10) The language of the article is relatively fluent and readable, with a reasonable academic writing standard, which is a relatively good academic paper. However, there are some presentation and punctuation errors in the article, which need further proofreading and revision by the author team. The writer also needs to turn some long and difficult sentences into short sentences that are easy to understand while reducing the use of passive voice sentences.

Reviewer 2 Report

Dear Authors,

1. In the introduction, the authors briefly introduce the background and significance of the research. However, I consider it necessary to highlight the connection between the research objectives and the research hypotheses.  

2. The literature review is unnecessary and should be deleted. Or the helpful information in the literature review can be moved to the first part.

3. There is no problem with the methodology.

4. The 4.7 discussion section should be included in a separate chapter. In addition, the discussion should consist of two sections, theoretical contribution, and practical implication.

5. Policy recommendations should be moved to the discussion section.

6. The manuscript lacks limitations and prospects. Please add it in the Conclusion section.

7. The manuscript quotes many Chinese papers, which I think should all be replaced by English articles, in line with the writing standards of English papers.

8. The biggest problem with this manuscript is that some parts contain too much irrelevant content or repetition. Therefore, it is recommended to delete a lot until 2-3 pages can shorten the manuscript.

Round 2

Reviewer 1 Report

Thank you for addressing my comments.

Reviewer 2 Report

Thank you for your revision.